# 3D Self-Supervised Methods for Medical Imaging

Aiham Taleb [1,*], Winfried Loetzsch [1,†], Noel Danz [1,†], Julius Severin [1,*], Thomas Gaertner [1,†], Benjamin Bergner [1,*], and Christoph Lippert [1,*]

[1]Digital Health & Machine Learning, Hasso-Plattner-Institute, Potsdam University, Germany
[*]{firstname.lastname}@hpi.de
[†]{firstname.lastname}@student.hpi.uni-potsdam.de

## Abstract

Self-supervised learning methods have witnessed a recent surge of interest after proving successful in multiple application fields. In this work, we leverage these techniques, and we propose 3D versions for five different self-supervised methods, in the form of proxy tasks. Our methods facilitate neural network feature learning from *unlabeled* 3D images, aiming to reduce the required cost for expert annotation. The developed algorithms are 3D Contrastive Predictive Coding, 3D Rotation prediction, 3D Jigsaw puzzles, Relative 3D patch location, and 3D Exemplar networks. Our experiments show that pretraining models with our 3D tasks yields more powerful semantic representations, and enables solving downstream tasks more accurately and efficiently, compared to training the models from scratch and to pretraining them on 2D slices. We demonstrate the effectiveness of our methods on three downstream tasks from the medical imaging domain: i) Brain Tumor Segmentation from 3D MRI, ii) Pancreas Tumor Segmentation from 3D CT, and iii) Diabetic Retinopathy Detection from 2D Fundus images. In each task, we assess the gains in data-efficiency, performance, and speed of convergence. Interestingly, we also find gains when transferring the learned representations, by our methods, from a large unlabeled 3D corpus to a small downstream-specific dataset. We achieve results competitive to state-of-the-art solutions at a fraction of the computational expense. We publish our implementations[1] for the developed algorithms (both 3D and 2D versions) as an open-source library, in an effort to allow other researchers to apply and extend our methods on their datasets.

## 1   Introduction

Due to technological advancements in 3D sensing, the need for machine learning-based algorithms that perform analysis tasks on 3D imaging data has grown rapidly in the past few years [1–3]. 3D imaging has numerous applications, such as in Robotic navigation, in CAD imaging, in Geology, and in Medical Imaging. While we focus on medical imaging as a test-bed for our proposed 3D algorithms in this work, we ensure their applicability to other 3D domains. Medical imaging plays a vital role in patient healthcare, as it aids in disease prevention, early detection, diagnosis, and treatment. Yet efforts to utilize advancements in machine learning algorithms are often hampered by the sheer expense of the expert annotation required [4]. Generating expert annotations of 3D medical images at scale is non-trivial, expensive, and time-consuming. Another related challenge in medical imaging is the relatively small sample sizes. This becomes more obvious when studying a particular disease, for instance. Also, gaining access to large-scale datasets is often difficult due to privacy concerns. Hence, scarcity of data and annotations are some of the main constraints for machine learning applications in medical imaging.

Several efforts have attempted to address these challenges, as they are common to other application fields of deep learning. A widely used technique is transfer learning, which aims to reuse the features of already trained neural networks on different, but related, target tasks. A common example is adapting the features from networks trained on ImageNet, which can be reused for other visual tasks, e.g. semantic segmentation. To some extent, transfer learning has made it easier to solve tasks with limited number of samples. However, as mentioned before, the medical domain is supervision-starved. Despite attempts to leverage ImageNet [5] features in the medical context [6–9], the difference in the distributions of natural and medical images is significant, i.e. generalizing across these domains is questionable and can suffer from dataset bias [10]. Recent analysis [11] has also found that such transfer learning offers limited performance gains, relative to the computational costs it incurs. Consequently, it is necessary to find better solutions for the aforementioned challenges.

A viable alternative is to employ self-supervised (unsupervised) methods, which proved successful in multiple domains recently. In these approaches, the supervisory signals are derived from the data. In general, we withhold some part of the data, and train the network to predict it. This prediction task defines a proxy loss, which encourages the model to learn semantic representations about the concepts in the data. Subsequently, this facilitates data-efficient fine-tuning on supervised downstream tasks, reducing significantly the burden of manual annotation. Despite the surge of interest in the machine learning community in self-supervised methods, only little work has been done to adopt these methods in the medical imaging domain. We believe that self-supervised learning is directly applicable in the medical context, and can offer cheaper solutions for the challenges faced by conventional supervised methods. Unlabelled medical images carry valuable information about organ structures, and self-supervision enables the models to derive notions about these structures with no additional annotation cost.

A particular aspect of most medical images, which received little attention by previous self-supervised methods, is their 3D nature [12]. The common paradigm is to cast 3D imaging tasks in 2D, by extracting slices along an arbitrary axis, e.g. the axial dimension. However, such tasks can substantially benefit from the full 3D spatial context, thus capturing rich anatomical information. We believe that relying on the 2D context to derive data representations from 3D images, in general, is a suboptimal solution, which compromises the performance on downstream tasks.

**Our contributions.** As a result, in this work, we propose five self-supervised tasks that utilize the full 3D spatial context, aiming to better adopt self-supervision in 3D imaging. The proposed tasks are: 3D Contrastive Predictive Coding, 3D Rotation prediction, 3D Jigsaw puzzles, Relative 3D patch location, and 3D Exemplar networks. These algorithms are inspired by their successful 2D counterparts, and to the best of our knowledge, most of these methods have never been extended to the 3D context, let alone applied to the medical domain. Several computational and methodological challenges arise when designing self-supervised tasks in 3D, due to the increased data dimensionality, which we address in our methods to ensure their efficiency. We perform extensive experiments using four datasets in three different downstream tasks, and we show that our 3D tasks result in rich data representations that improve data-efficiency and performance on three different downstream tasks. Finally, we publish the implementations of our 3D tasks, and also of their 2D versions, in order to allow other researchers to evaluate these methods on other imaging datasets.

## 2 Related work

In general, unsupervised representation learning can be formulated as learning an embedding space, in which data samples that are semantically similar are closer, and those that are different are far apart. The self-supervised family constructs such a representation space by creating a supervised proxy task from the data itself. Then, the embeddings that solve the proxy task will also be useful for other real-world downstream tasks. Several methods in this line of research have been developed recently, and they found applications in numerous fields [13]. In this work, we focus on methods that operate on images only.

Self-supervised methods differ in their core building block, i.e. the proxy task used to learn representations from unlabelled input data. A commonly used supervision source for proxy tasks is the spatial context from images, which was first inspired by the skip-gram Word2Vec [14] algorithm. This idea was generalized to images in [15], in which a visual representation is learned by predicting the position of an image patch relative to another. A similar work extended this patch-based approach to solve

Jigsaw Puzzles [16]. Other works have used different supervision sources, such as image colors [17], clustering [18], image rotation prediction [19], object saliency [20], and image reconstruction [21]. In recent works, Contrastive Predictive Coding (CPC) approaches [22, 23] advanced the results of self-supervised methods on multiple imaging benchmarks [24, 25]. These methods utilize the idea of contrastive learning in the latent space, similar to Noise Contrastive Estimation [26]. In 2D images, the model has to predict the latent representation for next (adjacent) image patches. Our work follows this line of research in the above works, however, our methods utilize the full 3D context.

While videos are rich with more types of supervisory signals [27–31], we discuss here a subset of these works that utilize 3D-CNNs to process input videos. In this context, 3D-CNNs are employed to simultaneously extract spatial features from each frame, and temporal features across multiple frames, which are typically stacked along the $3^{rd}$ (depth) dimension. The idea of exploiting 3D convolutions for videos was proposed in [32] for human action recognition, and was later extended to other applications [13]. In self-supervised learning, however, the number of pretext tasks that exploit this technique is limited. Kim *et al.* [33] proposed a task that extracts cubic puzzles of $2 \times 2 \times 1$, meaning that the $3^{rd}$ dimension is not actually utilized in puzzle creation. Jing *et al.* [34] extended the rotation prediction task [19] to videos, by simply stacking video frames along the depth dimension, however, this dimension is not employed in the design of their task as only spatial rotations are considered. Han *et al.* proposed a dense encoding of spatio-temporal frame blocks to predict future scene representations recurrently, in conjunction with a curriculum training scheme to extend the predicted future. Similarly, the depth dimension is not employed in this task. On the other hand, in our more general versions of 3D Jigsaw puzzles and 3D Rotation prediction, respectively, we exploit the depth ($3^{rd}$) dimension in the design of our tasks. For instance, we solve larger 3D puzzles up to $3 \times 3 \times 3$, and we also predict more rotations along all axes in the 3D space. Futhermore, in our 3D Contrastive Predictive Coding task, we predict patch representations along all 3 dimensions, scanning input volumes in a manner that resembles a pyramid. In general, we believe the different nature of the data, 3D volumetric scans vs. stacked video frames, influences the design of proxy tasks, i.e. the depth dimension has an actual semantic meaning in volumetric scans. Hence, we consider the whole 3D context when designing all of our methods, aiming to learn valuable anatomical information from unlabeled 3D volumetric scans.

In the medical context, self-supervision has found use-cases in diverse applications such as depth estimation in monocular endoscopy [35], robotic surgery [36], medical image registration [37], body part recognition [38], in disc degeneration using spinal MRIs [39], in cardiac image segmentation [40], body part regression for slice ordering [41], and medical instrument segmentation [42]. Spitzer *et al.* [43] sample 2D patches from a 3D brain, and predict the distance between these patches as a supervision signal. Tajbakhsh *et al.* [44] use orientation prediction from medical images as a proxy task. There are multiple other examples of self-supervised methods for medical imaging, such as [45–49]. While these attempts are a step forward for self-supervised learning in medical imaging, they have some limitations. First, as opposed to our work, many of these works make assumptions about input data, resulting in engineered solutions that hardly generalize to other target tasks. Second, none of the above works capture the complete spatial context available in 3-dimensional scans, i.e. they only operate on 2D/2.5D spatial context. In a more related work, Zhou *et al.* [50] extended image reconstruction techniques from 2D to 3D, and implemented multiple self-supervised tasks based on image-reconstruction. Zhuang *et al.* [51] and Zhu *et al.* [52] developed a proxy task that solves small 3D jigsaw puzzles. Their proposed puzzles were only limited to $2 \times 2 \times 2$ of puzzle complexity. Our version of 3D Jigsaw puzzles is able to efficiently solve larger puzzles, e.g. $3 \times 3 \times 3$, and outperforms their method's results on the downstream task of Brain tumor segmentation. In this paper, we continue this line of work, and develop five different algorithms for 3D data, whose nature and performance can accommodate more types of target medical applications.

## 3  Self-Supervised Methods

In this section, we discuss the formulations of our 3D self-supervised pretext tasks, all of which learn data representations from unlabeled samples (3D images), hence requiring no manual annotation effort in the self-supervised pretraining stage. Each task results in a pretrained encoder model $g_{enc}$ that can be fine-tuned in various downstream tasks, subsequently.

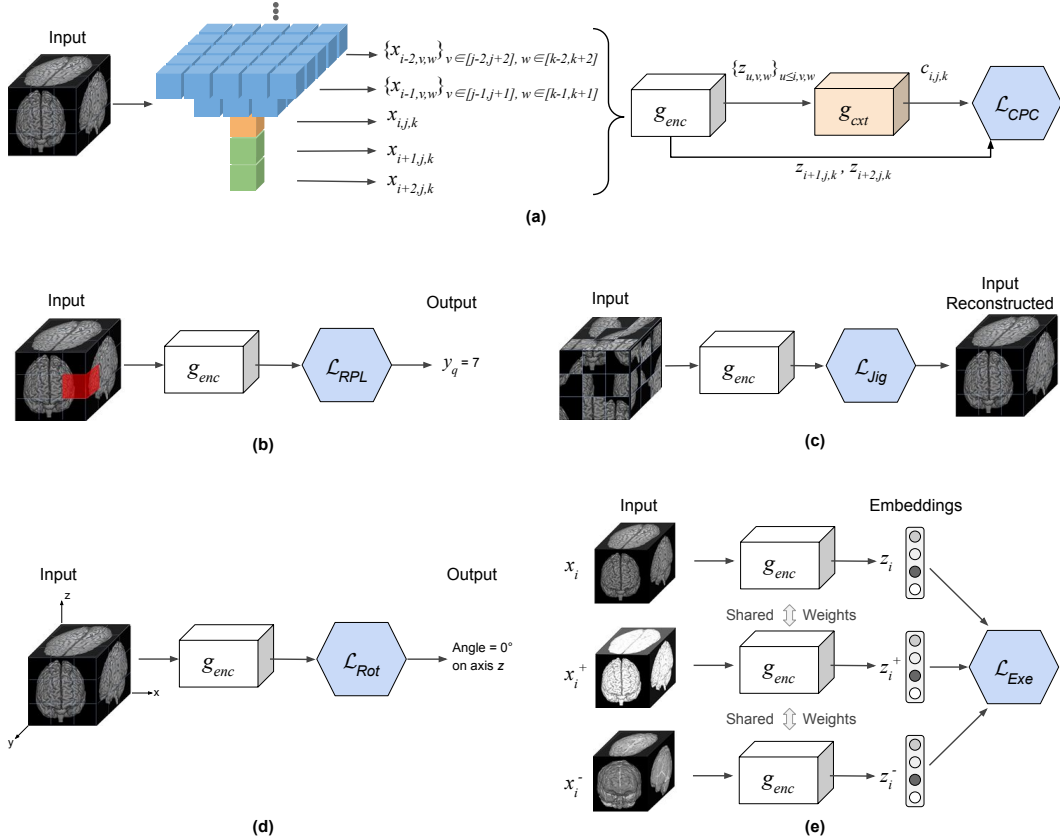

Figure 1: **(a)** 3D-CPC: each input image is split into 3D patches, and the latent representations $z_{i+1,j,k}, z_{i+2,j,k}$ of next patches $x_{i+1,j,k}, x_{i+2,j,k}$ (shown in green) are predicted using the context vector $c_{i,j,k}$. The considered context is the current patch $x_{i,j,k}$ (shown in orange), plus the above patches that form an inverted pyramid (shown in blue). **(b)** 3D-RPL: assuming a 3D grid of 27 patches ($3 \times 3 \times 3$), the model is trained to predict the location $y_q$ of the query patch $x_q$ (shown in red), relative to the central patch $x_c$ (whose location is 13). **(c)** 3D-Jig: by predicting the permutation applied to the 3D image when creating a $3 \times 3 \times 3$ puzzle, we are able to reconstruct the scrambled input. **(d)** 3D-Rot: the network is trained to predict the rotation degree (out of the 10 possible degrees) applied on input scans. **(e)** 3D-Exe: the network is trained with a triplet loss, which drives positive samples closer in the embedding space ($x_i^+$ to $x_i$), and the negative samples ($x_i^-$) farther apart.

## 3.1 3D Contrastive Predictive Coding (3D-CPC)

Following the contrastive learning idea, first proposed in [26], this universal unsupervised technique predicts the latent space for future (next or adjacent) samples. Recently, CPC found success in multiple application fields, e.g. its 1D version in audio signals [22], and its 2D versions in images [22, 23], and was able to bridge the gap between unsupervised and fully-supervised methods [24]. Our proposed CPC version generalizes this technique to 3D inputs, and defines a proxy task by cropping equally-sized and overlapping 3D patches from each input scan. Then, the encoder model $g_{enc}$ maps each input patch $x_{i,j,k}$ to its latent representation $z_{i,j,k} = g_{enc}(x_{i,j,k})$. Next, another model called the context network $g_{cxt}$ is used to summarize the latent vectors of the patches in the context of $x_{i,j,k}$, and produce its context vector $c_{i,j,k} = g_{cxt}(\{z_{u,v,w}\}_{u \leq i,v,w})$, where $\{z\}$ denotes a set of latent vectors. Finally, because $c_{i,j,k}$ captures the high level content of the context that corresponds to $x_{i,j,k}$, it allows for predicting the latent representations of next (adjacent) patches $z_{i+l,j,k}$, where $l \geq 0$. This prediction task is cast as an $N$-way classification problem by utilizing the InfoNCE loss [22], which takes its name from its ability to maximize the mutual information between $c_{i,j,k}$ and $z_{i+l,j,k}$. Here, the classes are the latent representations $\{z\}$ of the patches, among which is one *positive* representation, and the rest $N - 1$ are *negative*. Formally, the CPC loss can be written as

follows:

$$\mathcal{L}_{CPC} = -\sum_{i,j,k,l} \log p(z_{i+l,j,k} \mid \hat{z}_{i+l,j,k}, \{z_n\})$$

$$= -\sum_{i,j,k,l} \log \frac{\exp(\hat{z}_{i+l,j,k} z_{i+l,j,k})}{\exp(\hat{z}_{i+l,j,k} z_{i+l,j,k}) + \exp(\sum_n \hat{z}_{i+l,j,k} z_n)} \quad (1)$$

This loss corresponds to the categorical cross-entropy loss, which trains the model to recognize the correct representation $z_{i+l,j,k}$ among the list of negative representations $\{z_n\}$. These negative samples (3D patches) are chosen randomly from other locations in the input image. In practice, similar to the original NCE [26], this task is solved as a binary pairwise classification task.

It is noteworthy that the proposed 3D-CPC task, illustrated in Fig. 1 (a), allows employing any network architecture in the encoder $g_{enc}$ and the context $g_{cxt}$ networks. In our experiments, we follow [22] in using an autoregressive network using GRUs [53] for the context network $g_{cxt}$, however, masked convolutions can be a valid alternative [54]. In terms of what the 3D context of each patch $x_{i,j,k}$ includes, we follow the idea of an inverted pyramid neighborhood, which is inspired from [55, 56]. This context is chosen based on a tradeoff between computational cost and performance. Too large contexts (e.g. full surrounding of a patch) incur prohibitive computations and memory use. The inverted-pyramid context was an optimal tradeoff.

## 3.2 Relative 3D patch location (3D-RPL)

In this task, the spatial context in images is leveraged as a rich source of supervision, in order to learn semantic representations of the data. First proposed by Doersch *et al.* [15] for 2D images, this task inspired several works in self-supervision. In our 3D version, shown in Fig. 1 (b), we leverage the full 3D spatial context in the design of our task. From each input 3D image, a 3D grid of $N$ non-overlapping patches $\{x_i\}_{i \in \{1,..,N\}}$ is sampled at random locations. Then, the patch $x_c$ in the center of the grid is used as a reference, and a query patch $x_q$ is selected from the surrounding $N-1$ patches. Next, the location of $x_q$ relative to $x_c$ is used as the positive label $y_q$. This casts the task as an $N-1$-way classification problem, in which the locations of the remaining grid patches are used as the negative samples $\{y_n\}$. Formally, the cross-entropy loss in this task is written as:

$$\mathcal{L}_{RPL} = -\sum_{k=1}^{K} \log p(y_q \mid \hat{y}_q, \{y_n\}) \quad (2)$$

Where $K$ is the number of queries extracted from all samples. In order to prevent the model from solving this task quickly by finding shortcut solutions, e.g. edge continuity, we follow [15] in leaving random gaps (jitter) between neighboring 3D patches. More details in Appendix.

## 3.3 3D Jigsaw puzzle Solving (3D-Jig)

Deriving a Jigsaw puzzle grid from an input image, be it in 2D or 3D, and solving it can be viewed as an extension to the above patch-based RPL task. In our 3D Jigsaw puzzle task, which is inspired by its 2D counterpart [16] and illustrated in Fig. 1 (c), the puzzles are formed by sampling an $n \times n \times n$ grid of 3D patches. Then, these patches are shuffled according to an arbitrary permutation, selected from a set of predefined permutations. This set of permutations with size $P$ is chosen out of the $n^3!$ possible permutations, by following the Hamming distance based algorithm in [16] (details in Appendix), and each permutation is assigned an index $y_p \in \{1,..,P\}$. Therefore, the problem is cast as a $P$-way classification task, i.e., the model is trained to simply recognize the applied permutation index $p$, allowing us to solve the 3D puzzles in an efficient manner. Formally, we minimize the cross-entropy loss of $\mathcal{L}_{Jig}(y_p^k, \hat{y}_p^k)$, where $k \in \{1,..,K\}$ is an arbitrary 3D puzzle from the list of extracted $K$ puzzles. Similar to 3D-RPL, we use the trick of adding random jitter in 3D-Jig.

## 3.4 3D Rotation prediction (3D-Rot)

Originally proposed by Gidaris *et al.* [19], the rotation prediction task encourages the model to learn visual representations by simply predicting the angle by which the input image is rotated. The intuition behind this task is that for a model to successfully predict the angle of rotation, it needs

to capture sufficient semantic information about the object in the input image. In our 3D Rotation prediction task, 3D input images are rotated randomly by a random degree $r \in \{1, .., R\}$ out of the $R$ considered degrees. In this task, for simplicity, we consider the multiples of 90 degrees ($0°$, $90°$, $180°$, $270°$, along each axis of the 3D coordinate system $(x, y, z)$. There are 4 possible rotations *per axis*, amounting to 12 possible rotations. However, rotating input scans by $0°$ along the 3 axes will produce 3 identical versions of the original scan, hence, we consider 10 rotation degrees instead. Therefore, in this setting, this proxy task can be solved as a 10-way classification problem. Then, the model is tasked to predict the rotation degree (class), as shown in Fig. 1 (d). Formally, we minimize the cross-entropy loss $\mathcal{L}_{Rot}(r^k, \hat{r}^k)$, where $k \in \{1, .., K\}$ is an arbitrary rotated 3D image from the list of $K$ rotated images. It is noteworthy that we create multiple rotated versions for each 3D image.

### 3.5  3D Exemplar networks (3D-Exe)

The task of Exemplar networks, proposed by Dosovitskiy *et al.* [57], is one of the earliest methods in the self-supervised family. To derive supervision labels, it relies on image augmentation techniques, i.e. transformations. Assuming a training set $X = \{x_1, ...x_N\}$, and a set of $K$ image transformations $\mathcal{T} = \{T_1, ..T_K\}$, a new surrogate class $S_{x_i}$ is created by transforming each training sample $x_i \in X$, where $S_{x_i} = \mathcal{T}x_i = \{Tx_i \mid T \in \mathcal{T}\}$. Therefore, the task is cast as a regular classification task with a cross-entropy loss. However, this classification task becomes prohibitively expensive as the dataset size grows larger, as the number of classes grows accordingly. Thus, in our proposed 3D version of Exemplar networks, shown in Fig. 1 (e), we employ a different mechanism that relies on the triplet loss instead [58]. Formally, assuming $x_i$ is a random training sample and $z_i$ is its corresponding embedding vector, $x_i^+$ is a transformed version of $x_i$ (seen as a positive example) with an embedding $z_i^+$, and $x_i^-$ is a different sample from the dataset (seen as negative) with an embedding $z_i^-$. The triplet loss is written as follows:

$$\mathcal{L}_{Exe} = \frac{1}{N_T} \sum_{i=1}^{N_T} \max\{0, D(z_i, z_i^+) - D(z_i, z_i^-) + \alpha\} \tag{3}$$

where $D(.)$ is a pairwise distance function, for which we use the $L_2$ distance, following [59]. $\alpha$ is a margin (gap) that is enforced between positive and negative pairs, which we set to $1$. The triplet loss enforces $D(z_i, z_i^-) > D(z_i, z_i^+)$, i.e. the transformed versions of the same sample (positive samples) to come closer to each other in the learned embedding space, and farther away from other (negative) samples. Replacing the triplet loss with a contrastive loss [26] is possible in this method, and has been found to improve learned representations from natural images [24]. In addition, the learned representations by Exemplar can be affected by the negatives sampling strategy. The simple option is to sample from within the same batch, however, it is also possible to sample from the whole dataset. The latter choice is computationally more expensive, but is expected to improve the learned representations, as it makes the task harder. It is noteworthy that we apply the following 3D transformations: random flipping along an arbitrary axis, random rotation along an arbitrary axis, random brightness and contrast, and random zooming.

## 4  Experimental Results

In this section, we present the evaluation results of our methods, which we assess the quality of their learned representations by fine-tuning them on three downstream tasks. In each task, we analyze the obtained gains in data-efficiency, performance, and speed of convergence. In addition, each task aims to demonstrate a certain use-case for our methods. We follow the commonly used evaluation protocols for self-supervised methods in each of these tasks. The chosen tasks are:

- Brain Tumor Segmentation from 3D MRI (Subsection 4.1): in which we study the possibility for transfer learning from a different unlabeled 3D corpus, following [60].

- Pancreas Tumor Segmentation from 3D CT (Subsection 4.2): to demonstrate how to use the same unlabeled dataset, following the data-efficient evaluation protocol in [23].

- Diabetic Retinopathy Detection from 2D Fundus Images (Subsection 4.3): to showcase our implementations for the 2D versions of our methods, following [23]. Here, we also evaluate pretraining on a different large corpus, then fine-tuning on the downstream dataset.

We provide additional details about architectures, training procedures, the effect of augmentation in Exemplar, and how we initialize decoders for segmentation tasks in the Appendix.

## 4.1 Brain Tumor Segmentation Results

In this task, we evaluate our methods by fine-tuning the learned representations on the Multimodal Brain Tumor Segmentation (BraTS) 2018 [61, 62] benchmark. Before that, we pretrain our models on brain MRI data from the UK Biobank [63] (UKB) corpus, which contains roughly $22K$ 3D scans. Due to this large number of unlabeled scans, UKB is suitable for unsupervised pretraining. The BraTS dataset contains annotated MRI scans for $285$ training and $66$ validation cases. We fine-tune on BraTS' training set, and evaluate on its validation set. Following the official BraTS challenge, we report Dice scores for the Whole Tumor (WT), Tumor Core (TC), and Enhanced Tumor (ET) tasks. The Dice score (F1-Score) is twice the area of overlap between two segmentation masks divided by the total number of pixels in both. In order to assess the quality of the learned representations by our 3D proxy tasks, we compare to the following baselines:

- Training from scratch: the first sensible baseline for any self-supervised method, in general, is the same model trained on the downstream task when initialized from random weights. Comparing to this baseline provides insights about the benefits of self-supervised pretraining.

- Training on 2D slices: this baseline aims to quantitatively show how our proposal to operate on the 3D context benefits the learned representations, compared to 2D methods.

- Supervised pretraining: this baseline uses automatic segmentation labels from FSL-FAST [64], which include masks for three brain tissues.

- Baselines from the BraTS challenge: we compare to the methods [65–68], which all use a single model with an architecture similar to ours, i.e. 3D U-Net [69].

**Discussion.** We first assess the gains in data-efficiency in this task. To quantify these gains, we measure the segmentation performance at different sample sizes. We randomly select subsets of patients at 10%, 25%, 50%, and 100% of the full dataset size, and we fine-tune our models on these subsets. Here, we compare to the baselines listed above. As shown in Fig. 2, our 3D methods outperform the baseline model trained from scratch by a large margin when using few training samples, and behaves similarly as the number of labeled samples increases. The low-data regime case at 5% suggests the potential for generic unsupervised features, and highlights the huge gains in data-efficiency. Also, the proposed 3D versions considerably outperform their 2D counterparts, which are trained on slices extracted from the 3D images. We also measure how our methods affect the final brain tumor segmentation performance, in Table 1. All our methods outperform the baseline trained from scratch as well as their 2D counterparts, confirming the benefits of pretraining with our 3D tasks on downstream performance. We also achieve comparable results to baselines from the BraTS challenge, and we outperform these baselines in some cases, e.g. our 3D-RPL method outperforms all baselines in terms of ET and TC dice scores. Also, our model pretrained with 3D-Exemplar, with fewer downstream training epochs, matches the result of Isensee *et al.* [65] in terms of WT dice score, which is one of the top results on the BraTS 2018 challenge. In comparison to the supervised baseline using automatic FAST labels, we find that our results are comparable, outperforming this baseline in some cases. Our results in this downstream task also demonstrate the generalization ability of our 3D tasks across different domains. This is result is significant, because medical datasets are supervision-starved, e.g. images may be collected as part of clinical routine, but much fewer (high-quality) labels are produced, due to annotation costs.

## 4.2 Pancreas Tumor Segmentation Results

In this downstream task, we evaluate our models on 3D CT scans of Pancreas tumor from the medical decathlon benchmarks [70]. The Pancreas dataset contains annotated CT scans for $420$ cases. Each scan in this dataset contains 3 different classes: pancreas (class 1), tumor (class 2), and background (class 0). To measure the performance on this benchmark, two dice scores are computed for classes 1 and 2. In this task, we pretrain using our proposed 3D tasks on pancreas scans *without* their annotation masks. Then, we fine-tune the obtained models on subsets of annotated data to assess the gains in both data-efficiency and performance. Finally, we also compare to the baseline model trained from scratch and to 2D models, similar to the previous downstream task. Fig. 3 demonstrates the gains

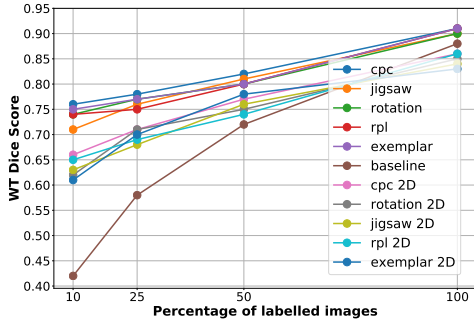

Figure 2: Data-efficient segmentation results in BraTS. With less labeled data, the supervised baseline (brown) fails to generalize, as opposed to our methods. Also, the proposed 3D methods outperform all 2D counterparts.

Table 1: BraTS segmentation results

| Model | ET | WT | TC |
|---|---|---|---|
| 3D-From scratch | 76.38 | 87.82 | 83.11 |
| 3D Supervised | 78.88 | 90.11 | 84.92 |
| 2D-CPC | 76.60 | 86.27 | 82.41 |
| 2D-RPL | 77.53 | 87.91 | 82.56 |
| 2D-Jigsaw | 76.12 | 86.28 | 83.26 |
| 2D-Rotation | 76.60 | 88.78 | 82.41 |
| 2D-Exemplar | 75.22 | 84.82 | 81.87 |
| Popli *et al.* [66] | 74.39 | 89.41 | 82.48 |
| Baid *et al.* [67] | 74.80 | 87.80 | 82.66 |
| Chandra *et al.* [68] | 74.06 | 87.19 | 79.89 |
| Isensee *et al.* [65] | 80.36 | **90.80** | 84.32 |
| 3D-CPC | 80.83 | 89.88 | 85.11 |
| 3D-RPL | **81.28** | 90.71 | **86.12** |
| 3D-Jigsaw | 79.66 | 89.20 | 82.52 |
| 3D-Rotation | 80.21 | 89.63 | 84.75 |
| 3D-Exemplar | 79.46 | **90.80** | 83.87 |

when fine-tuning our models on 5%, 10%, 50%, and 100% of the full data size. The results obtained by our 3D methods also outperform the baselines in this task with a margin when using only few training samples, e.g. 5% and 10% cases. Another significant benefit offered by pretraining with our methods is the speed of convergence on downstream tasks. As demonstrated in Fig 5, when training on the full pancreas dataset, within the first 20 epochs only, our models achieve much higher performances compared to the "from scratch" baseline. We should note that we evaluate this task on a held-out labeled subset of the Pancreas dataset that was not used for pretraining nor fine-tuning. We provide the full list of experimental results for this task in Appendix.

## 4.3  Diabetic Retinopathy Results

As part of our work, we also provide implementations for the 2D versions of the developed self-supervised methods. We showcase these implementations on the Diabetic Retinopathy 2019 Kaggle challenge 4.3. This dataset contains roughly $5590$ Fundus 2D images, each of which was rated by a clinician on a severity scale of $0$ to $4$. These levels define a classification task. In order to evaluate our tasks on this benchmark, we pretrain all the 2D versions of our methods using 2D Fundus images from UK Biobank [63]. The retinopathy data in UK Biobank contains $170K$ images. We then fine-tune the obtained models on Kaggle data, meaning performing transfer learning. We also compare the obtained results with this transfer learning protocol to those obtained with the data-efficient evaluation protocol in [23], i.e. pretraining on the same Kaggle dataset and fine-tuning on subsets of it. To assess the gains in data-efficiency, we fine-tune the obtained models on subsets of labelled Kaggle data, shown in Fig. 4. It is noteworthy that pretraining on UKB produces results that outperform those obtained when pretraining on the same Kaggle dataset. This confirms the benefits of transfer learning from a large corpus to a smaller one using our methods. Gains in speed of convergence are also shown in Fig. 6. In this 2D task, we achieve results consistent with the other downstream tasks, presented before. We should point out that we evaluate with 5-fold cross validation on this 2D dataset. The metric used in task, as in the Kaggle challenge, is the Quadratic Weighted Kappa, which measures the agreement between two ratings. Its values vary from random (0) to complete (1) agreement, and if there is less agreement than chance it may become negative.

## 5  Conclusion

In this work, we asked whether designing 3D self-supervised tasks could benefit the learned representations from unlabeled 3D images, and found that it indeed greatly improves their downstream performance, especially when fine-tuned on only small amounts of labeled 3D data. We demonstrate the obtained gains by our proposed 3D algorithms in data-efficiency, performance, and speed of convergence on three different downstream tasks. Our 3D tasks outperform their 2D counterparts, hence supporting our proposal of utilizing the 3D spatial context in the design of self-supervised tasks, when operating on 3D domains. What is more, our results, particularly in the low-data regime,

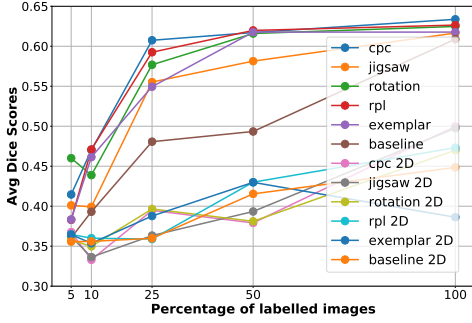

Figure 3: Data-efficient segmentation results in Pancreas. With less labeled data, the supervised baseline (brown) fails to generalize, as opposed to our methods. Also, the proposed 3D methods outperform all 2D counterparts

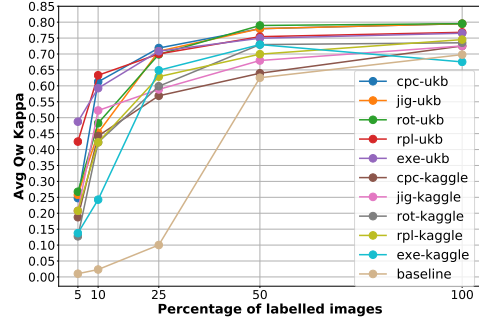

Figure 4: Data-efficient classification in Diabetic Retinopathy. With less labels, the supervised baseline (brown) fails to generalize, as opposed to pretrained models. This result is consistent with the other downstream tasks

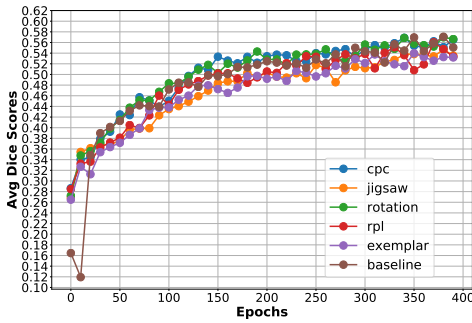

Figure 5: Speed of convergence in Pancreas segmentation. Our models converge faster than the baseline (brown)

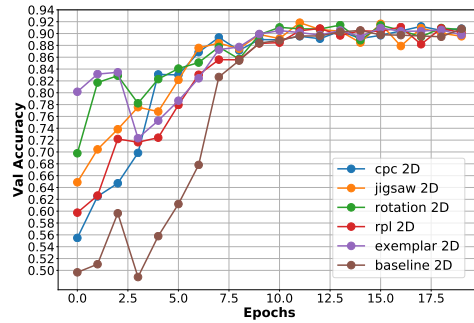

Figure 6: Speed of convergence in Retinopathy classifcation. Our models also converge faster in this task

demonstrate the possibility to reduce the manual annotation effort required in the medical imaging domain, where data and annotation scarcity is an obstacle. Furthermore, we observe performance gains when pretraining our methods on a large unlabeled corpus, and fine-tuning them on a different smaller downstream-specific dataset. This result suggests alternatives for transfer learning from Imagenet features, which can be substantially different from the medical domain. Finally, we open source our implementations for all 3D methods (and also their 2D versions), and we publish them to help other researchers apply our methods on other medical imaging tasks. This work is only a first step toward creating a set of methods that facilitate self-supervised learning research for 3D data, e.g. medical scans. We believe there is room for improvement along this line, such as designing new 3D proxy tasks, evaluating different architectural options, and including other data modalities (e.g. text) in conjunction with images/scans.

## Broader Impact

Due to technological advancements in 3D data sensing, and to the growing number of its applications, the attention to machine learning algorithms that perform analysis tasks on such data has grown rapidly in the past few years. As mentioned before, 3D imaging has multitude of applications [2], such as in Robotics, in CAD imaging, in Geology, and in Medical Imaging. In this work, we developed multiple 3D Deep Learning algorithms, and evaluated them on multiple 3D medical imaging benchmarks. Our focus on medical imaging is motivated by the pressing demand for automatic (and instant) analysis systems, that may aid the medical community.

Medical imaging plays an important role in patient healthcare, as it aids in disease prevention, early detection, diagnosis, and treatment. With the continuous digitization of medical images, the hope that physicians and radiologists are able to instantly analyze them with Machine Learning algorithms is slowly shaping as a reality. Achieving this has become more critical recently, as the number of patients which contracted with a novel Coronavirus, called COVID-19, reached a high record. Radiography images provide a rich and a quick diagnosis tool, because other types of tests, e.g. RT-PCR which is an RNA/DNA based test, have low sensitivity and may require hours/days of processing [71]. Therefore, as imaging allows such instant insights into human body organs, it receives growing attention from both machine learning and medical communities.

Yet efforts to leverage advancements in machine learning, particularly the supervised algorithms, are often hampered by the sheer expense of expert annotation required [4]. Generating expert annotations of patient data at scale is non-trivial, expensive, and time-consuming, especially for 3D medical scans. Even current semi-automatic software tools fail to sufficiently address this challenge. Consequently, it is necessary to rely on annotation-efficient machine learning algorithms, such as self-supervised (unsupervised) approaches for representation learning from unlabelled data. Our work aims to provide the necessary tools for 3D image analysis, in general, and to aid physicians and radiologists in their diagnostic tasks from 3D scans, in particular. And as the main consequence of this work, the developed methods can help reduce the effort and cost of annotation required by these practitioners. In the larger goal of leveraging Machine Learning for good, our work is only a small step toward achieving this goal for patient healthcare.

## Acknowledgments and Disclosure of Funding

This research has been supported by funding from the German Federal Ministry of Education and Research (BMBF) in the project KI-LAB-ITSE (project number 01|S19066). This research has been conducted using the UK Biobank Resource.

## Footnotes

[1]https://github.com/HealthML/self-supervised-3d-tasks

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
