[Supplementary Material]

# Appendix for: 3D Self-Supervised Methods for Medical Imaging

Aiham Taleb [1,*], Winfried Loetzsch [1,†], Noel Danz [1,†], Julius Severin [1,*], Thomas Gaertner [1,†], Benjamin Bergner [1,*], and Christoph Lippert [1,*]

[1]Digital Health & Machine Learning, Hasso-Plattner-Institute, Potsdam University, Germany
[*]{firstname.lastname}@hpi.de
[†]{firstname.lastname}@student.hpi.uni-potsdam.de

## 1 Implementation and training details for all tasks

It is noteworthy that our attached implementations are flexible enough to allow for evaluating several types of network architectures for encoders, decoders, and classifiers. We also provide implementations for multiple losses, augmentation techniques, and evaluation metrics. More information can be found in the `README.md` file in our attached code-base. We rely on `tensorflow v2.1` [1] with `Keras API` in our implementations. Below, we provide the training details we used in implementing our 3D self-supervised tasks (and their 2D counterparts), and when fine-tuning them in subsequent downstream tasks.

**Architecture details.** For all 3D encoders $g_{enc}$, which are pretrained with our 3D self-supervised tasks and later fine-tuned on 3D segmentation tasks, we use a 3D U-Net [2]-based encoder (the downward path), which consists of five levels of residual convolutional blocks. The numbers of filters in these blocks are $32, 64, 128, 256, 512$, respectively. The U-Net decoder (the upward path) is added in the downstream fine-tuning stage, and it includes five levels of deconvolutional blocks with skip connections from the U-Net encoder blocks. For the 2D encoders, we use a standard Densenet-121 [3] architecture, which is fine-tuned later on 2D classification tasks. When training our 3D self-supervised tasks, we follow [4] in adding nonlinear transformations (a hidden layer with ReLU activation) before the final classification layers. These classification layers are removed when fine-tuning the resulting encoders $g_{enc}$ in downstream tasks.

**Optimization details.** In all self-supervised and downstream tasks, we use Adam [5] optimizer to train the models. The initial learning rate we use is $0.001$ in 3D self-supervised tasks, $0.00001$ in 3D segmentation tasks, $0.0005$ in 2D self-supervised tasks, and $0.00005$ in 2D classification tasks. When we fine-tune our pretrained encoders in subsequent downstream tasks, we follow a warm-up procedure inspired from [6] by keeping the encoder weights frozen for a number of initial warm-up epochs while the network decoders or classifiers are trained. These warm-up epochs are 5 in 2D classification tasks, and 25 epochs in 3D segmentation tasks. The alternative options we evaluated were: 1) fine-tuning the encoder directly with a randomly initialized decoder, 2) keeping the encoder frozen throughout the training procedure. And the 3$^\text{rd}$ option we followed in the end was the hybrid approach of warm-up epochs described above, as it provided a performance boost over the other alternatives. For segmentation tasks, in particular, where a decoder is used in the architecture, these warm-up epochs prove indispensable. Otherwise, training the whole model with a randomly initialized decoder, while the encoder is not frozen, may harm the encoder representations.

**Input preprocessing.** For all input scans, we perform the following preprocessing steps:

- In self-supervised pretraining using 3D scans, we find the boundaries of the brain or the pancreas along each axis, and then we crop the remaining empty parts from the scan. This

step reduces the amount of empty background voxels, as they might confuse patch-based self-supervised methods with no additional semantic information. This step is not performed when fine-tuning on 3D downstream tasks.

- Then, we resize each 3D image from BraTS or Pancreas to a unified resolution of $128 \times 128 \times 128$, and to the resolution $224 \times 224$ for 2D images from Diabetic Retinopathy.
- Then, each image's intensity values are normalized by scaling them to the range $[0, 1]$.

**Processing multimodal inputs.** In the first downstream task of brain tumor segmentation with 3D multimodal MRI, we pretrain using the UK Biobank [7] corpus, as mentioned earlier. Brain scans obtained from UKB contain 2 MRI modalities (T1 and T2-Flair), which are co-registered. This allows us to stack these 2 modalities as color channels in each input sample, similar to RGB channels. This form of early fusion [8] of MRI modalities is common when they are registered, and is a practical solution for combining all information that exist in these modalities. However, as mentioned earlier, we use the BraTS [9, 10] dataset for fine-tuning, and each scan consists of 4 different MRI modalities, as opposed to only 2 in UKB that is used for pretraining. This difference only affects the input layer of the pretrained encoder, as fine-tuning on an incompatible number of input channels causes this process of fine-tuning to fail. We resolve this issue by duplicating (copying) the weights of *only* the pretrained input layer. This minor modification only adds a few additional parameters to the input layer, but allows us to leverage its weights. The other alternative for this solution would have been to discard the weights of this input layer, and initialize the rest of the model layers from pretrained models normally. But we believe our solution for this issue takes advantage of any useful information encoded in these weights. This multimodal inputs problem does not occur in the other downstream tasks, as the inputs include only one modality/channel.

**Task specific training details.**

- **3D-CPC and 3D-Exe**: we use latent representation code size of 1024 in these tasks.
- **3D-Jig and 3D-RPL**: We split the input 3D images into $3 \times 3 \times 3$ patches in this task. We apply a random jitter of 3 pixels per side (axis).
- **Patch-based tasks (3D-CPC, 3D-RPL, 3D-Jig)**: each extracted patch is represented using an embedding vector of size 64.
- **3D-Exe**: the $\alpha$ value used for the triplet loss is $1.0$.
- **3D-Jig**: the complexity of the Jigsaw puzzle solving task relies on the number of permutations used in generating the puzzles, i.e. the more permutations used, the harder the task is to solve. We follow the Hamming distance-based algorithm from [11] in sampling the permutations for this task. However, in our 3D puzzles task, we sample permutations that are more complex with 27 different entries. This algorithm works as follows: we sample a subset of 1000 permutations which are selected based on their Hamming distance, i.e., the number of different tile locations between 2 permutations. When generating permutations, we ensure that the average Hamming distance across permutations is kept as high as possible. This results in a set of permutations (classes) that are as far as possible from each other.

**Augmentation in Exemplar.** As mentioned earlier, we apply the following 3D transformations in Exemplar: random flipping along an arbitrary axis, random rotation along an arbitrary axis, random brightness and contrast, and random zooming. These augmentations are utilized to produce the positive samples. We vary the percentages of applying these augmentations using these factors: $\alpha = 0.5$ for random rotations, $\beta = 0.5$ for color distortions (brightness and contrast), and $\gamma = 0.2$ for random zooming. When trying to omit a certain augmentation from the list above, we observe a drop in downstream performance. This is consistent with the findings of [4]. However, performing such transformations for high percentages is time-consuming, hence the reduced rates to $50\%$. Conducting a more thorough analysis of what *types* of augmentations are desirable is a future work.

# 2 Detailed experimental results

(a) CPC 3D vs. baseline

(b) RPL 3D vs. baseline

(c) Jigsaw 3D vs. baseline

(d) Rotation 3D vs. baseline

(e) Exemplar 3D vs. baseline

Figure 1: Pancreas segmentation: Detailed data-efficiency results per method (blue) vs. the supervised baseline (orange). Our methods consistently outperform the baseline in low-data cases

(a) CPC 3D vs. baseline

(b) RPL 3D vs. baseline

(c) Jigsaw 3D vs. baseline

(d) Rotation 3D vs. baseline

(e) Exemplar 3D vs. baseline

Figure 2: Pancreas segmentation: Detailed speed of convergence results per method (blue) vs. the supervised baseline (orange). This benefit of our methods helps achieve high results using only few epochs

(a) CPC 2D vs. baseline  (b) RPL 2D vs. baseline  (c) Jigsaw 2D vs. baseline

(d) Rotation 2D vs. baseline  (e) Exemplar 2D vs. baseline

Figure 3: Retinopathy detection: Detailed data-efficiency results per method (blue) vs. the supervised baseline (orange). Our methods consistently outperform the baseline in low-data cases

(a) CPC 2D vs. baseline  (b) RPL 2D vs. baseline  (c) Jigsaw 2D vs. baseline

(d) Rotation 2D vs. baseline  (e) Exemplar 2D vs. baseline

Figure 4: Retinopathy detection: Detailed speed of convergence results per method (blue) vs. the supervised baseline (orange). This benefit of our methods helps achieve high results using only few epochs