[Reviews · NeurIPS 2020]

Review 1

Summary and Contributions: The authors extend five known auxiliary tasks in computer vision to improve the performance of 3D segmentation deep learning models. Authors report results evaluated on 4 medical imaging datasets, demonstrating that including surrogate tasks in a self-learning setting improves the performance of the baselines.

Strengths: - Results improve the baselines. - The paper is easy to read.

Weaknesses: - There is neither technical contribution of this paper nor novel interesting insights. Auxiliary tasks are known to improve the representation learning of deep models and boost the performance on mainstream tasks. Extending 5 well known auxiliary tasks to 3D cannot be considered as a significant technical contribution. - Evaluation is also weak. Since there are many open questions by using these auxiliary tasks, authors could further investigate how the affect the performance of main models. E.g., is the combination of multiple auxiliary tasks improving the performance when having only single tasks?

Correctness: - Claims are somehow correct. Authors claim that they perform extensive experiments. Nevertheless, authors just evaluated the auxiliary tasks on 4 datasets, simply comparing the baselines with baselines + auxiliary tasks in 2D and 3D. To me, this is not what I understand as extensive experiments. - Methodology is correct.

Clarity: The paper is easy to read.

Relation to Prior Work: Prior work is correctly discussed. Since this paper extends prior works to 3D, there is no much to discuss about the differences from previous literature.

Reproducibility: Yes

Additional Feedback: After reading other reviewers' concerns and the answers from the authors I keep my initial score. I am not convinced about the arguments from authors regarding the motivation of their work, i.e., self-training has been mainly obtained in ImageNet. There exist a whole body of literature that employs self-training, by means of auxiliary tasks, for medical imaging tasks, including both 2D and 3D scenarios (A short list of references that were quickly found is provided at the end). Both the related work and the author's answer show that they overlooked relevant prior work, as they mention that all previous work focus on either 2D or 2.5D, and most work is only performed in ImageNet. I kindly disagree with this statement, and weakens the contribution of their work. Finally, while I agree that extending CPC to 3D is not straightforward, the rest of the tasks is trivial. Thus, this contribution alone is insufficient for this conference. ===== References ====== -Zhuang et al. Self-supervised feature learning for 3D medical images by playing a rubik’s cube. MICCAI 2019 (pp. 420-428). Springer, Cham. - Zhou et al. Models genesis: Generic autodidactic models for 3D medical image analysis. MICCAI 2019 Oct 13 (pp. 384-393). - Zhu et al Rubik’s Cube+: A Self-supervised Feature Learning Framework for 3D Medical Image Analysis. Medical Image Analysis. 2020 Jun 6:101746. - Spitzer et al. Improving cytoarchitectonic segmentation of human brain areas with self-supervised siamese networks.MICCAI 2018. - Tajbakhsh et al. Surrogate supervision for medical image analysis: Effective deep learning from limited quantities of labeled data. IEEE ISBI 2019 Apr 8 (pp. 1251-1255). IEEE. - Bai et al. Self-supervised learning for cardiac mr image segmentation by anatomical position prediction. MICCAI 2019 Oct 13 (pp. 541-549).


Review 2

Summary and Contributions: The paper proposes 3D equivalents of popular self-supervised learning algorithms for 2D images, and applies these to medical imaging tasks. Contributions include (1) Proposed instantations of CPC, RPL, Jigsaw, Rotation, and Exemplar. Significance: Low-Medium (2) They apply these algorithms to data efficient brain tumor segmentation. Significance: Very High (2) They apply these algorithms to data efficient pancreas tumor segmentation. Significance: Medium (2) They apply these algorithms to data efficient diabetic retinopathy. Significance: Medium

Strengths: The authors extend 5 self-supervised algorithms to 3D, and show that they all work well (compared to training without additional labelled data). This coverage of multiple methods is admirable. The biggest strength of the paper is the successful application to medical imaging tasks. Most development in self-supervision is performed on ImageNet. Getting these methods to work well in another domain, and on 3D data, is an important step forward for these methods. The application is, of course, very important. I am particularly impressed by the Brain Tumor experiments, where the authors exploit an auxiliary unlabelled corpus (UKB) for the self-supervised learning and show significant gains on the BraTS dataset. The gains are significant even when using all of the labels available in BraTS. The paper is clearly written, details are given in the appendix, and code is available, so I feel reasonably confident that the results are reliable.

Weaknesses: The actual methodology itself does not feel groundbreaking. The extensions from 2D to 3D seem relatively natural. From the paper, I did not feel that there was any major extension or change to the methods required to extend from 2D to 3D, other than tuning, of course (perhaps I missed a critical detail?). I do not consider the method being “simple” a major weakness though. If there were any pitfalls encountered when shifting from 2D to 3D, it would be useful to see a discussion of what “obvious” ideas did not work. The pancreas tumor and retinopathy experiments are less exciting than the brain tumor experiments. In both cases unsupervised data is created artificially by discarding labels from a labelled dataset. Such an artificial setup is unfortunately common in the literature. If the authors could provide a good justification that in these specific applications it is realistic that unlabelled data would be available *from the exact same distribution* as the labelled data, then I would be more excited by these experiments. I am not familiar with the UKB dataset, but after a search, it seems that there are some labels available from longitudinal studies. It would be a nice baseline to first pretrain on these labels, then apply transfer learning to BraTS. There is no discussion of the computational requirements for 3D self-supervision. The hardware used, flops spent compared to 2D counterparts, etc. For pancreas tumor segmentation, their method gets near SOTA (Isensee et al.). SOTA is not required for publication. However, could you discuss why their method is marginally better (presumably it does not use 3D self-supervision) -- could it be a bigger network, auxiliary data, just more tuning, etc? Fig. 4, why does Exemplar get worse from 50% to 100% data? Is this trend real or noise? If it is real, then somehow the tuning of the algorithm seems incorrect, surely more data can't hurt? If it is noise, then error bars are needed, since all of the curves seem to lie within the implied noise level.

Correctness: The methods look correct, and the experimental protocol looks solid. Details are provided in the Appendix, as well as code.

Clarity: The paper is clearly written, no complaints.

Relation to Prior Work: I am not familiar enough with medical literature to guarantee 100% recall of all relevant literature. However, the literature presented is clearly framed in the context of this paper.

Reproducibility: Yes

Additional Feedback: I am currently learning towards acceptance. I could be convinced to increase my score further if you could address the comments in the Weaknesses section. In addition, could you discuss - How much does data augmentation matter, in particular for Exemplar. Recent extensions of Exemplar, like SimCLR show big gains from tuning the data augmentation correctly. - On Exemplar, in standard ImageNet evaluations Exemplar-based methods far outperform others. Yet, in your experiments, Exemplar is not the best. Could this be due to the excessive domain-specific tuning required for this algorithm. I think such a discussion would be valuable for the community.


Review 3

Summary and Contributions: This paper reviews the latest progress of different self-supervised learning strategies and adapt these strategies to 3D medical image analysis. Extensive experiments are conducted to demonstrate the effectiveness of 3D-based self-supervised learning.

Strengths: 1) This paper might be impactful in the field of medical image analysis since self-supervised pre-training has been proved to be quite helpful for reducing annotation burden in natural images. But for medical image analysis, a comprehensive evaluation and benchmarks are missing. The authors provide code on different medical imaging tasks which can be beneficial for a lot of further studies in this direction. 2) The writing is good.

Weaknesses: The novelty is modest. All 5 different strategies, i.e.,3D Contrastive Predictive Coding, 3D Rotation prediction, 3D Jigsaw puzzles, Relative 3D patch location, and 3D Exemplar networks are all existing techniques. This paper seems to only adapt these methods to 3D space and include them in their library. In this sense, this paper may be more like a technical report rather than an academic paper.

Correctness: claims and method correct; empirical methodology correct

Clarity: yes

Relation to Prior Work: yes

Reproducibility: Yes

Additional Feedback: From the technical side, I would be probably be interested to see how to better modify these self-supervised methods for medical image analysis by taking advantage of some specific prior knowledge in the medical domain. Then the current version can be served as a general baseline. ========================= Update: After seeing other reviewers' comments and the author's feedback, I now tend to be borderline towards this paper (probably still slightly towards accept, but wouldn't be upset if this paper gets rejected). Thereby I lowered my grade from 7 to 6. And here is my side: The pros and cons for this paper seems obvious---on one hand, the paper summarizes the usage of 5 different pretext tasks for unsupervised pretraining in medical image analysis, which could be of great impact; on the other hand, the technical contribution does seem limited since all 5 pretext tasks are existing methods, and there is not in-depth discussion such as what is learned in different pretext tasks, etc. Now this might make the paper look more like a technical report rather than an academic paper. However, I still do believe this paper might have potential impact and strongly encourage the authors to well address reviewers' comments in updated versions.


Review 4

Summary and Contributions: The authors propose use-cases of commonly used self-supervision techniques in the context of analysing 3D medical scans for tasks such as semantic segmentation and scan classification. The experimental evaluations are performed on public benchmarks such as BRATS Challenge (tumour segmentation) and Kaggle DR fundus image classification, plus the authors release their implementations. I think the authors have done a great work on benchmarking such metric learning approaches in an extensive experimentation framework; however, the summary of the related work and proposed novelties are far-below the expectations for such a conference. Therefore, I would recommend revision of this article by taking into account the feedback below and resubmission to another venue.

Strengths: Extensive evaluation on public benchmarks.

Weaknesses: It’s unclear to me why the authors would like to utilise, and benchmark five different self-supervision approaches given that the current SOTA is set by contrastive learning approaches. Additionally, the recent work show that contrastive learning alone is sufficient to achieve far-better results than methods that rely on contrastive predictive coding. In short, I would clarify the motivation for proposing all these alternatives as it dilutes the main message and confuses the reader. [1] “A Simple Framework for Contrastive Learning of Visual Representations.” Ting Chen, Simon Kornblith, Mohammad Norouzi, Geoffrey Hinton [2] “Momentum Contrast for Unsupervised Visual Representation Learning”. Kaiming He, Haoqi Fan, Yuxin Wu, Saining Xie, Ross Girshick More importantly, it’s unclear what the potential technical challenges are in extending such well-established techniques to 3D imaging data. Maybe the authors could refer to 2D+T longitudinal modelling literature where the community has tried to learn embeddings from video datasets e.g. contrastive learning or metric learning literature. [3] “Video Representation Learning by Dense Predictive Coding”, Tengda Han, Weidi Xie, Andrew Zisserman Lastly, it’s interesting that the authors aim to solve a structured-prediction task (segmentation) with the self-learnt embeddings. Do you think that a methodology developed for a global prediction task can be applied as an off-the-shelf method to a structured prediction task where spatial details and texture are concerned? Can we expect that embeddings do preserve such granular information as minimisation of contrastive object would not necessarily require the network to learn such information to identify nearest neighbours? I think it would be better to shed more light on such details and give a clear motivation instead of solely applying it on a healthcare dataset and reporting performance results.

Correctness: -

Clarity: The paper reads very well.

Relation to Prior Work: Some recent technical references are missing which are highlighted in the comments above. If the authors are interested in including recent work from the healthcare space, I could suggest the following as well: "Models Genesis: Generic Autodidactic Models for 3D Medical Image Analysis"

Reproducibility: Yes

Additional Feedback: Post-rebuttal feedback: "Please refer to R2-1 and novelty comments for a discussion about technical challenges. Moreover, as explained in lines (95-111), in contrast to 2D+T methods, our methods exploit the whole 3D context (including the depth dimension)." I am not sure why spatio-temporal models trained with SSL objective would not capture spatial context. I am aware that there is probably more redundancy in sequence data in comparison to a volumetric scans; however, the underlying methodology is almost identical. "Solving a segmentation task with the self-learnt prediction embeddings. What about spatial/texture details?" Is the model required to extract texture information to minimise a SSL objective? (e.g. jigsaw) If so how do we know that it's actually learning that? I think it's important to realise the difference between structured and global prediction tasks, where in the former models need to capture fine spatial details and object boundaries present in images. However, I am not sure even the best performing SSL approach (e.g. cpc) is able to capture that since it's not encouraged in the training objective. Indeed, this is one of the reasons why we see these methods applied mainly only on imagenet and cifar classification benchmarks. Overall, I do not see any clear clinical and technical justification for the design choices in this work. The presented results might be perhaps of interest to some researcher in the healthcare space which can be discussed in a dedicated workshop.

[Author Response · NeurIPS 2020]

We thank the reviewers for their supportive and insightful comments. We address their questions/concerns below.

**Comments about novelty** (*R1*: Limited technical contribution, *R3*: The novelty is modest):
As noted by *R2&R3*, existing results for self-supervised methods have mainly been obtained on ImageNet. We extend these methods to 3D medical imaging, where labels are expensive to obtain, by pretraining on a large unlabeled corpus (UK Biobank) or on images from the same dataset. Furthermore, as explained in *R2*-1, we propose extensions that work in 3D contexts, e.g. for CPC, which was not trivial. Moreover, generalizing a concept from lower- to higher-dimensions is common in the literature (see [1] and lines 95-126 in our paper), and can offer insights for novel applications.

*R2:* **1-** "The extensions from 2D to 3D seem relatively natural. . . pitfalls encountered when shifting from 2D to 3D."
Extending CPC to 3D was not straightforward. In 1D, the future values are predicted based on history. In 2D, the prediction is performed row- and column-wise, i.e. solving many 1D tasks. In our experiments, similarly small contexts yielded poor results in 3D. Too large contexts (e.g. full surrounding of a patch) incurred prohibitive computations and memory use. The inverted-pyramid context was an optimal tradeoff. We will include a comparison of these variants.
**2-** In pancreas tumor and retinopathy experiments, "unsupervised data is created artificially by discarding labels.."
Medical datasets are supervision-starved (lines 27-33), e.g. images may be collected as part of clinical routine, but much fewer (high-quality) labels are produced, due to annotation costs. However, we agree that a transfer learning setting is more significant, as it leverages additional data from different distributions. Hence, we pretrained on Retinopathy data from the UK Biobank (170K images), and fine-tuned on Kaggle data (5K images). Transfer learning yielded gains (in Qw-Kappa), in 24/25 settings (see table). We plan to include pancreas-tumor segmentation into this evaluation.

| Model / (% of data) | CPC | | | | | RPL | | | | | Jigsaw | | | | | Rotation | | | | | Exemplar | | | | |
|---|---|---|---|---|---|---|---|---|---|---|---|---|---|---|---|---|---|---|---|---|---|---|---|---|---|
| | 5 | 10 | 25 | 50 | 100 | 5 | 10 | 25 | 50 | 100 | 5 | 10 | 25 | 50 | 100 | 5 | 10 | 25 | 50 | 100 | 5 | 10 | 25 | 50 | 100 |
| Pretrained (UKB) | 24 | 61 | 71 | 77 | 79 | 42 | 63 | 70 | 75 | 76 | 25 | 45 | 70 | 77 | 79 | 26 | 48 | 69 | 78 | 79 | 48 | 59 | 70 | 74 | 76 |
| Baseline (Kaggle) | 18 | 44 | 56 | 63 | 72 | 20 | 42 | 62 | 69 | 74 | 20 | 52 | 58 | 67 | 72 | 12 | 42 | 59 | 72 | 73 | 13 | 24 | 64 | 72 | 67 |

**3-** UK Biobank baseline pretrained on longitudinal segmentation labels, and transfer learning to BraTS (100% labels). The longitudinal labels are for fMRI. However, we added an experiment based on automatic labels from FSL-FAST, which include masks for three brain tissues. Our results in Tab.1 (paper) are comparable to this baseline (table below).

| Model / BraTS Metrics | ET | WT | TC |
|---|---|---|---|
| Pretraining on FAST masks (UKB) | 78.88 | 90.11 | 84.92 |

**4-** Discussion of the computational requirements (hardware used, flops spent, etc). We will add these to the final version.
**5-** For brain-tumor segmentation, our methods get near Isensee et al.'s. Discuss why their method is marginally better. Isensee et al. use more training data, a larger U-Net, and post-processing. Our 3D-RPL is comparable (lines 259-263).
**6-** Fig. 4, why Exemplar gets worse at 100%? Is this trend real or noise? If it is noise, then error bars are needed. We believe this drop at 100% of the data is caused by noise, and hence will add error bars to the final version.
**7-** How much does data augmentation matter, in particular for Exemplar. Recently, SimCLR shows big gains. Our findings are consistent with SimCLR, i.e. combined data augmentations in Exemplar improve learned representations. However, the types of augmentations may differ. An analysis about this will be added to the final version.
**8-** On ImageNet, Exemplar-based methods outperform others. Yet, in our experiments, Exemplar is not the best.. Exemplar-based methods can be affected by: training loss (contrastive vs. triplet), domain-specific tuning, negative sampling (batch vs. dataset), . . . We discuss this in the final version. Also, implementing a 3D SimCLR is a future work.

*R3:* **1-** How to modify these methods by taking advantage of some specific prior knowledge in the medical domain.
We aim to develop novel methods that utilize data-locality in 3D. Thank you for the suggestion.

*R4:* **1-** The motivation of five self-supervision approaches given that the SOTA is set by contrastive learning approaches.
As accurately noted by *R2,R3*, all previous SOTA is set on ImageNet, and it is hard to generalize such results to different contexts (2D natural vs. 3D medical images). We plan to extend contrastive approaches to 3D contexts in the future.
**2-** Potential technical challenges, and a comparison to 2D+T methods on video inputs.
Please refer to *R2*-1 and *novelty comments* for a discussion about technical challenges. Moreover, as explained in lines (95-111), in contrast to 2D+T methods, our methods exploit the whole 3D context (including the depth dimension).
**3-** Solving a segmentation task with the self-learnt prediction embeddings. What about spatial/texture details?
The applicability of our methods to several tasks is a benefit we had in mind. We agree that segmentation tasks require learning more details, however, our results in Fig.2&Fig.3 confirm that pretraining the encoder only is able to capture generic data representations, similar to other self-supervised methods [2]. This enforces the decoder network to capture these spatial and texture details during fine-tuning. We will add an analysis to the final version.
**4-** Some recent technical references are missing. SimCLR is ref. (24) in our paper. We will add the others, thanks.

**References**

[1] Carl Vondrick, Abhinav Shrivastava, Alireza Fathi, Sergio Guadarrama, and Kevin Murphy. Tracking emerges by colorizing videos. In *Proceedings of the European Conference on Computer Vision (ECCV)*, September 2018.
[2] Yonglong Tian, Dilip Krishnan, and Phillip Isola. Contrastive multiview coding. *ArXiv*, abs/1906.05849, 2019.


[Meta-Review · NeurIPS 2020]

Applying self-supervision methods to the medical domain is interesting to the community, and adaptation of the methods from 2D to 3D domain is also of value in the machine learning community. There are numerous self-supervision works on medical images mentioned by R1 post rebuttal. The authors are encouraged to include those in the paper, and properly place their paper in the context of previous self-supervision work in the medical imaging domain.